# A New Pest Detection Method Based on Improved YOLOv5m

**DOI:** 10.3390/insects14010054

**Published:** 2023-01-05

**Authors:** Min Dai, Md Mehedi Hassan Dorjoy, Hong Miao, Shanwen Zhang

**Affiliations:** College of Mechanical Engineering, Yangzhou University, Yangzhou 225127, China

**Keywords:** deep learning, convolutional neural networks, pest detection, YOLOv5m

## Abstract

**Simple Summary:**

Insect pests can damage crops and food production, causing problems for farmers. The detection of plant pests is essential for ensuring the excellent productivity of plants and food. Traditional methods for pest detection are generally time-consuming and inefficient. There has been a lot of use of deep learning for detecting plant pests in recent years. YOLOv5 is one of the most effective deep learning algorithms used for object detection. A new pest detection method with higher accuracy based on a deep convolutional neural network (CNN) is proposed in this paper. Experimental results on the pest dataset indicate that the proposed method performs well and can achieve high precision and robustness for recognizing plant pests. The proposed method is more effective and can detect pests precisely with higher accuracy.

**Abstract:**

Pest detection in plants is essential for ensuring high productivity. Convolutional neural networks (CNN)-based deep learning advancements recently have made it possible for researchers to increase object detection accuracy. In this study, pest detection in plants with higher accuracy is proposed by an improved YOLOv5m-based method. First, the SWin Transformer (SWinTR) and Transformer (C3TR) mechanisms are introduced into the YOLOv5m network so that they can capture more global features and can increase the receptive field. Then, in the backbone, ResSPP is considered to make the network extract more features. Furthermore, the global features of the feature map are extracted in the feature fusion phase and forwarded to the detection phase via a modification of the three output necks C3 into SWinTR. Finally, WConcat is added to the fusion feature, which increases the feature fusion capability of the network. Experimental results demonstrate that the improved YOLOv5m achieved 95.7% precision rate, 93.1% recall rate, 94.38% *F*_1_ score, and 96.4% Mean Average Precision (*mAP*). Meanwhile, the proposed model is significantly better than the original YOLOv3, YOLOv4, and YOLOv5m models. The improved YOLOv5m model shows greater robustness and effectiveness in detecting pests, and it could more precisely detect different pests from the dataset.

## 1. Introduction

A vast number of pests are threatening crops. People have always used pesticides to get rid of pests in agriculture. However, there are other ways, such as mechanical control, trapping, and recently developed biological controls. Timely detection of plant pests is one of the significant challenges in agriculture. In recent years, the problem of stored grain pests has become more and more serious. Insect pests are responsible for 20% of the annual crop losses that occur across the world [1]. With the advancement of computer vision technology, machine learning and deep learning have been used in the agricultural pest identification field [2,3,4,5,6].

Traditional machine learning is mainly based on computer vision to extract pest texture, color, shape, and other features, using a support vector machine (SVM) [7], K-nearest neighbor (KNN) [8], and other algorithms to detect target pests [9,10,11,12,13]. The machine learning-based agricultural pest detection technology requires a complex pest image feature design. However, in the wild environment, the background of crops’ pests is complex and changeable, and light dramatically affects the shooting. It is difficult to manually change pests’ color and shape characteristics from similar ones. Therefore, it is challenging to apply machine-learning techniques to fulfill the need for the autonomous monitoring of pests.

Deep learning has been used to identify plant pests and diseases. With deep learning, plant diseases and pest detection tasks can be unified into an end-to-end feature extraction approach, which has excellent development prospects and significant potential, compared to traditional image processing methods, which deal with plant diseases and pest detection tasks in several steps and links [4]. Automatic pest detection research in agriculture is an essential research subject, since it can demonstrate the benefits of monitoring large areas and detecting pests automatically when they appear on a plant. Pest insects can be detected and monitored automatically with the aid of sensors. Infrared, audio, and image-based sensors are used to identify pests, along with recent advances such as machine learning [14]. At present, the commonly used deep learning target detection network is the Faster RCNN algorithm (faster region with the convolutional neural network, Faster RCNN) [15], Single shot multi-box detector (SSD) [16,17], and YOLO algorithm (You only look once) [18,19,20]. In particular, YOLO models such as YOLOv2, YOLOv3, and YOLOv4 are widely used to detect objects.

In recent years, YOLO-based target detection algorithms have been successfully applied in pest detection. For instance, YOLOv5 was more efficient at detecting insect pests with higher accuracy and about 2.5 times faster when the background images were simple, such as the Baidu AI insect detection dataset [21]. To improve the accuracy of insect identification, the YOLOv3-AFF network with the adaptive feature fusion was employed to reuse features at various scales, and it achieved 72.10% accuracy on the Pest24 dataset [22]. Researchers have proposed targeted network model structures based on different application scenarios to identify pests in real-world scenarios. Liu and Wang used image pyramids to optimize the feature layer of YOLOv3 for the recognition task of tomato pests in practical scenarios, and the recognition accuracy reached to 92.39% [23]. Liu et al. addressed an enhanced YOLOv4 object detection technique to improve the accuracy of tomato pests [24]. Zhang et al. introduced an improved YOLOv5 network to detect unopened cotton bolls in the field precisely, which can effectively assist farmers to take effective approaches in time and reduce crop losses and increase production [25]. Li et al. proposed a YOLO-DPD model that combines the residual feature augmentation and attention mechanism modules to detect and identify three diseases and pests on the rice canopy [26]. Guo et al. presented an automatic monitoring scheme to detect vegetable insect pests using a monitoring device and a YOLO-SIP detector, and the average accuracy was 84.22% [27]. Li et al. developed a jute pest identification model YOLO-JD for detecting pests from images and YOLO-JD achieved the Mean Average Precision (*mAP*) of 96.63% [28]. Chen et al. applied YOLOv4 to detect and localize scale pests in the images and it achieved the highest classification accuracy with it [29]. Therefore, YOLO has shown the great advantages in the field of pest detection.

In response to the above problem, we investigate pests’ detection from the dataset using improved YOLOv5m object detection architecture for the recognition of plant pests. In this paper, four YOLO models are employed: YOLOv3, YOLOv4, YOLOv5m, and an improved version of YOLOv5m. The model’s accuracy, effectiveness, and robustness need to be improved to obtain a better result. Therefore, it is a technical problem that needs to be solved to combine the YOLO model with the actual detection task. In order to achieve pest detection more accurately, the following improvements are made based on the YOLOv5m network structure:

(1) To capture more global features and increase the receptive field, a SWin Transformer (named SWinTR) and new Transformer (named C3TR) mechanisms are added to the YOLOv5m network.

(2) ResSPP is considered to increase the network’s capability of extracting features in the backbone. In addition, WConcat is added to the fusion feature, which increases the feature fusion capability of the network.

(3) In the feature fusion phase, the feature map’s global features are extracted and sent to the detection phase using a modified version of the three output necks C3 into SWinTR. The SWin Transformer is provided to the three detection headers of the network in the output section to further enhance the Mean Average Precision (*mAP*) of the model by making better use of the global feature information.

For convenience, the main abbreviations in this paper are summarized in Table 1**.** The remaining sections of this paper are organized as follows: Section 2 introduces the materials and methods. An improved YOLOv5m-based method is presented in this section. Experimental results and analysis are addressed in Section 3. Section 4 summaries the conclusions and future works.

## 2. Materials and Methods

### 2.1. Overview

Figure 1 illustrates the overall development of our project in this paper based on the deep convolutional neural network models for pest detection described as follows. The first step is constructing a pest dataset that contains train, validation and test data. Those data are labeled by labelImg, and after that the dataset is used for deep learning models. The created pest dataset is used for YOLOv3, YOLOv4, YOLOv5m and an improved model. For training and testing, the same data set is used for all these models. The next step is training and testing the model to obtain the output results. The last step is the deep model recommendation, indicating which model performs better.

### 2.2. Pest Dataset

The pest dataset used in this paper comes mainly from public datasets, direct field observation, and online resources. Figure 2 illustrates the sample images, including ten categories selected from the pest dataset. To test the validity of the algorithms, the pest of each category has same size in the test images and pests are visible under different angles. The images from the pest dataset are saved in JPG format with various resolutions. Various environments, image resolutions, and image kinds are represented in the dataset. In order to increase the samples’ diversity and complexity, the pests are photographed from different angles and directions. The pest images are randomly cropped to 640 × 640 due to the trade-off between performance and computational complexity in model training. In total, we created a dataset with 1309 images shared of each category pest, of which 1178 are used for training and 131 for validation and test shared together. In addition, the images are annotated using free software called labelImg. YOLO format is used for the annotations, which includes plant pests and bounding box coordinates.

### 2.3. Model Design

#### 2.3.1. The Structure of YOLOv5m Model

YOLOv5 is one of the most effective deep learning algorithms used for object detection, and YOLOv5 is based on YOLOv1-YOLOv4. YOLOv5 includes the YOLOv5s, YOLOv5m, YOLO5l, and YOLOv5x variants, each of which has a unique width and depth. Figure 3 demonstrates the structure of the YOLOv5m network. It has three primary components: (1) Backbone, (2) Neck, and (3) Head (output). Each component is described as below:

(1) Backbone: Backbone networks extract feature maps of different sizes from input images by pooling and multiple convolutions. Four Conv layers of feature maps were created in the backbone network, as observed in Figure 3. There are different sizes of feature maps such as small, medium, and large.

(2) Neck: A group of network layers that integrate image features and forward them to the prediction layer. The neck network fuses the feature maps of several levels to gather more contextual information and lessen information loss. The feature pyramid structures of FPN and PAN are utilized in the fusion process. It is a top-down and bottom-up feature fusion path for FPN and PAN. FPN and PAN jointly enhance the feature fusion capability of the neck network. The feature fusion layers generate three sizes of new feature maps, such as small, medium, and large.

(3) Head (output): Head is to predict image features, construct bounding boxes, and then predict categories. The output network part detects objects based on these new feature maps.

In particular, the backbone of the YOLOv5m network is a crucial feature extraction part, which includes the Focus module, Conv module, C3 module, and SPP module. The Focus module in the backbone performs slice operation on the images and concatenates them in order to improve feature extraction during down-sampling. Down-sampling is a method of reducing the size of images, which is used to reduce the dimension of features and retain effective information. The Conv module represents the convolutional layer, batch normalization layer, and SiLU activation function combination. There are two kinds of C3 in the YOLOv5m model framework. The backbone network uses one, and the neck uses another one. The structural differences between the two types of C3 networks are minimal. Backbone C3 networks contain residual units, whereas neck C3 networks replace residual units with the Conv module. The C3 module contains three standard convolutional layers and multiple bottleneck modules. There are two branches in the C3 module. One branch employs multiple bottleneck stacks and three standard convolution layers. In contrast, the other branch passes through only one basic convolution layer, finally concatenating the two branches. The bottleneck module can reduce training parameters and the computation of the model, which solves gradient explosion and gradient disappearance in deep networks. Hence each layer may extract relevant information and increase the model’s learning ability. Figure 3 shows that input X passes through a Conv1 module to halve the number of input channels. As a result, three different convolution kernels are combined for maximum pooling to produce an output with the same size and number of channels. When Concat splicing is performed, the number of channels is adjusted so that it is equivalent to Conv2. After maximum pooling down-sampling of three different convolution kernels, the number of output channels remains the same, which helps further splicing. SPP primarily aims to expand the receiving field, fuse information from feature maps of different scales, and complete the feature fusion process. The up-sample is used for up-sampling layer fusion in the nearest node. Slicing is performed by concatenating layers. Finally, the output is the Head of YOLOv5m, which predicts using three different-sized feature maps. It allows the model to manage objects of various sizes, such as small, medium, and large.

#### 2.3.2. The Method of Improved YOLOv5m Model

The task of this research is pest detection. The goal is to achieve higher accuracy and faster detection with the smallest possible network structure through improved YOLOv5m. As shown in Figure 4, an improved YOLOv5m target detection algorithm that can be used for pest detection is proposed. Significant changes are made to the model’s backbone. First, to capture more global features and increase the receptive field, the SWin Transformer (SWinTR) and Transformer (C3TR) are introduced into the YOLOv5m network. Then, ResSPP is considered to make the network extract more features. In addition, in the feature fusion part, two modifications are made: SWinTR replaces C3 at the network’s three output detection headers and incorporates Concat with weights.

(1) SWinTR and C3TR

The Transformer has achieved excellent results in natural language processing. The Transformer evolved initially as part of the natural language processing (NLP) field [30]. With Transformer, features can be adaptively aggregated from a global view using self-attention mechanisms rather than convolution operations, which extract features from a local fixed field [31]. The form of standard self-attention operation can be formulated (see Equations (1) and (2)) as follows [32]:(1)$$Q=f_{Q}(X),K=f_{K}(X),V=f_{V}(X)$$
(2)$$Attention(Q,K,V)=soft\max\left(\frac{Q^{T}K}{\sqrt{d}}\right)V$$
where *X* is the input feature map; $f_{Q}\left(\cdot \right),f_{K}\left(\cdot \right),f_{V}\left(\cdot \right)$ are the linear transformation functions; d is the channel dimension; Q,K,V are abbreviations of query, key, and weight value, respectively. We can calculate the dot products of the query with all keys, divide by $\sqrt{d}$, and apply a SoftMax function to obtain the weights on the values. After that, the attention function on a set of queries is calculated simultaneously, packing together into a matrix *Q*. Keys and values are also packed into matrices *K* and *V*, respectively.

By comparing the Bottleneck layer of the C3 module in Figure 3, the C3TR replaces the original Bottleneck block in C3 with a Transformer block, as shown in Figure 4. The standard Transformer layer is employed to obtain global information from the last block of the Backbone (C3TR block in Figure 4). The Transformer Encoder consists of two main parts: the Multi-head Self-Attention Mechanism (*MSA*). The *MSA* consists of multiple sets of attention. The query value (*Q*), key value (*K*), and weight value (*V*) obtained by normalization containing global feature information of different subspaces are updated and stitched together for linear projection. Finally, the features are outputted nonlinearly by the MLP to improve the expression of the self-attentive mechanism to capture contextual information and reduce the loss of global information.

The Transformer’s self-attention mechanism can effectively pick up global features. The Swin transformer builds a hierarchical feature map that introduces the Transformer into computer vision without more computation, and the image size has linear computational complexity [33]. The SWin transformer mechanism is added to improve the model’s receptive field and ability to extract features. The SWin Transformer is used in the three detection headers of the network, and the global features of the features are fully integrated before the network output to improve the model’s *mAP*. First, the window-based multi-head self-attention layer is observed. The feature map is divided into several small windows. Each window contains *M* × *M* patches (4 × 4 in the picture but usually 7 × 7 in reality).

Assume that the map size is *h* × *w*, then the window number is *h/M* × *w/M*. When self-attention is applied, it only does attention locally inside each window. The advantage of the window-based multi-head cell potential layer is that it can reduce computational costs for the input image size from quadratic to linear complexity. Figure 5 shows how to calculate the computational cost of *MSA,* as the input is of size *h* × *w* × *c*. It will multiply with matrixes wQ,wK, and wV, respectively. The wQ, wK, and wV are all of *c* × *c*; therefore, these three multiplication costs are 3*hwc*^2^. When *Q* multiplies *K*, the cost is (*hw*)^2^*c*; another (*hw*)^2^*c* is produced by *A* multiplied *V* after getting the dot product of the *A* and *V*. Here, we need one more projection there that the cost will be *hwc*^2^. Hence, *MSA* cost is calculated below in total (see Equation (3)) [32]:(3)$$\Omega (MSA)=4hwC^{2}+2{\left(hw\right)}^{2}c$$

Furthermore, hw is considered to replace the original self-attention with *h/M* × *w/M* to obtain the cost of each window and time of the window number to obtain the *W-MSA*. It can be observed that the complexity degrades from quadratic hw square to linear hw. Thus, *W-MSA* cost is calculated as below (see in Equation (4)) [32]:(4)$$\Omega (W-MSA)=4hwC^{2}+2M^{2}hwc$$

As shown in Figure 5, the Swin Transformer makes a hierarchical feature map. This means it can be used in computer vision without extra computation. At the same time, the size of the image has linear complexity in terms of computation. Figure 6a shows how the Swin transformer builds a hierarchical representation by merging small patches in a deeper transformer layer, starting with the smallest patches. Hierarchical feature maps help the Swin transformer model make accurate predictions.

Figure 6b shows that the shift of the window partition between successive self-attention layers is one of the essential features of the Swin transformer’s design. A disadvantage of the window-based multi-head self-attention layer is that it cannot exchange information between different windows. There is a way to solve this use of the shifted window to encourage information exchange. Moving pictures to the top left equals moving the windows to the bottom right, then patches that are not in the same window before are now shifted into one window. Thus, local self-attention can be conducted. It is an excellent solution, but one issue is dealing with incomplete segments, because batch computation needs to take the input with the same dimension. A straightforward way is to pad each incomplete segment to a complete window set, increasing the window number. For example, two by two is going to be three by three.

Based on the above self-attention mechanism, the SWin Transformer is introduced into the YOLOv5m network structure as one of its layers (Figure 4). SWinTR is used to replace the third layer of the backbone. It is done to increase the network’s receptive field and make it possible for the backbone to obtain more global features. SWinTR replaces the original C3 at the network’s three output detection headers during the feature fusion phase. Hence, it is possible to obtain the global semantic information of the feature map.

(2) ResSPP:

The SPP module is used to achieve feature fusion, which can be observed in Figure 3. In the improved model, the residual edge is added to YOLOv5m and it can perform better with SPP, which is named ResSPP. The purpose of using ResSPP is to allow our network to extract more features so that the residue edge can be added to make it easier for the network to do backpropagation without disappearing. The change is being made due to the residue network, which itself has the capability. ResSPP is part of the backbone modification. For the ResSPP structure, Figure 4 shows that input x passes through SPP, which is the same SPP structure as observed in Figure 3. Our goal is to output the SPP and then do a convolution. Upon performing the convolution, the output will be obtained. Therefore, the output will be added to the input again. The output is where the changes are made.

(3) WConcat:

WConcat is a feature fusion of weights. The splicing of two feature maps in this section should be more than just a simple concatenation of the two feature maps. There will be different weights assigned to each of these two feature maps. They are fused after giving weight to the feature map. The activation function is applied, and the output is obtained. Thus, the WConcat formula is given below (see Equations (5) and (6)) [34]:(5)$$output=Conv(\delta (\sum_{i=0}^{1}w_{i}x_{i}))$$
(6)$$w_{i}=\frac{w_{i}^{\prime }}{\sum_{j=0}^{1}w_{j}^{\prime }+\varepsilon }$$
where *x_i_* is the input and *i* equals 0 to 1. *x*_0_ is a feature map; *x*_1_ is the other feature map. δ represents the activation function. *w_i_* will store different weights, respectively, and then add them together. $w_{i}^{\prime }$ is before normalization. ε prevents the denominator from being 0.

The parameter can be obtained from training. The purpose of the parameter is to train the network and allow the network to suggest how much weight to give to *x*_0_ and *x*_1_. $w_{0}^{\prime }$ and $w_{1}^{\prime }$ are both obtained through network training. $w_{i}$ should be equal to the $w_{i}$ input by the network, divided by the summation symbol, and the summation symbol is *j* that equals 0 to 1. $w_{i}$ equals to it because of normalization. For example, if the network outputs two features, two weights, and one is $w_{0}^{\prime }$, and one is $w_{1}^{\prime }$, then the formula does not use $w_{0}^{\prime }$ and $w_{1}^{\prime }$ directly; it just uses $w_{0}$ and $w_{1}$. Thus, their relationship is described as below (see Equations (7) and (8)):(7)$$w_{0}=\frac{w_{0}^{\prime }}{w_{0}^{\prime }+w_{1}^{\prime }+\varepsilon }$$
(8)$$w_{1}=\frac{w_{1}^{\prime }}{w_{0}^{\prime }+w_{1}^{\prime }+\varepsilon }$$

If the minimal ε will not be considered, $w_{0}$ and $w_{1}$ should add up to 1, which is normalized. In short, the essential formula is the above formula (see Equation (5)). The core idea is to give weight to different feature maps. The feature map is obtained from the training information; thus, normalization is added here.

Figure 7 shows the overview of the WConcat splicing method. In order to obtain the output, feature maps x_0_ and x_1_ are spliced after W weight, then non-linearized with the Relu activation function, and then adjusted by convolution.

## 3. Experimental Results and Analyses

### 3.1. Experimental Setting

The experiments in this paper were carried out on a Windows 10 operating system, processor information: Intel (R) Core (TM) i5-9300H CPU @ 2.40 GHz, graphics card information: GeForce GTX 1650. PyTorch version 1.9.0 is used to build the network model in the experiment, and python version 3.8.10 is used. In addition, we used CUDA version 10.2 (Computer Unified Device Architecture) for GPU (Graphics Processing Unit) acceleration to improve computer graphics computing. The related parameter settings are described in Table 2.

### 3.2. Performance Evaluation

To evaluate the effectiveness of the improved YOLOv5m model on pest detection, *Precision*, and *Recall*, Mean Average Precision (*mAP*) and *F*_1_ score are used as evaluation indicators. The mathematical expressions are introduced as follows (see Equations (9)–(12)).
(9)$$Precision=\frac{TP}{TP+FP}$$
(10)$$Recall=\frac{TP}{TP+FN}$$
(11)$$mAP=\frac{1}{Q}\sum_{q=Q}AP(q)$$
(12)$$F_{1}\;score=2\times \frac{Precision\times Recall}{Precision+Recall}$$
where *TP* represents the number of positive samples anticipated to be positive; in contrast, *FP* represents the number of negative samples predicted to be positive. *FN* indicates the number of positive samples that are projected to be negative.

*mAP* stands for mean *Precision*, which is an effective and more accurate way to interpret model effects. It is used to evaluate the network’s detection accuracy, and its value indicates the effectiveness of network detection. As for *mAP*, the *P-R* curve is adopted to determine the *AP* value. Here, *P* reflects how many actual instances are in the positive case results, and *R* reflects how many actual instances are recalled. Average *Precision* is referred to as *AP*. *AP* represents the average value of all *Precision* obtained under all possible values of the recall rate. When the intersection over union (*IoU*) threshold is 0.5, the higher the *mAP*@0.5 is, the better the model’s performance.

*Precision* and *Recall* results are combined to determine the *F*_1_ score. *Precision* and *Recall* increase when the *F*_1_ score increases, and vice versa. The model cannot be considered effective if it has high *Precision* but poor *Recall*. The *F*_1_ score indicates the robustness of the model. The higher the *F*_1_ score value, the better the robustness.

### 3.3. Results and Discussion

To investigate the model’s capability to detect pest targets as well as its practical application ability, we analyzed the performance indicators of the proposed model, including the *Precision*, *Recall*, and *mAP*. The experimental results are shown in Figure 8.

As shown in Figure 8, the *Precision* and *Recall* rate quickly tends to the steady values as epochs increase, and they can almost reach 1.0 after 85 epochs. It has both excellent *Precision* and *Recall*, which could demonstrate the perfect performance of the model. Moreover, it shows a steady increase in *mAP*@0.5 in the first 50, and its increase rate is fast. The graph remains relatively stable after 85 epochs. Thus, the model can achieve high values of *Precision*, *Recall*, and *mAP*@0.5.

According to the above evaluation indicators, we further evaluate the pest detection performance by employing four different YOLO models, including YOLOv3, YOLOv4, YOLOv5m, and our proposed YOLOv5m. The comparison results of different YOLO models regarding performance indicators (*F*_1_ score, *P* curve, and *R* curve) over confidence are shown in Figure 9.

Figure 9 illustrates that when the confidence threshold increases then the precision goes up, and the recall goes down. Specifically speaking, when the confidence level is 0.337, the *F*_1_ score of original YOLOv3 reaches its maximum value of 0.89. The maximum *Precision* value is 1.00 when the confidence is 0.751. Maximum *Recall* is 0.97 when the confidence level is 0.0. In addition, the *F*_1_ score of original YOLOv4 achieves its maximum value of 0.91 when the confidence level is 0.509. The maximum *Precision* value is 1.00 when the confidence is 0.705. Maximum *Recall* is 0.97 when the confidence level is 0.0. Furthermore, the *F*_1_ score of the original YOLOv5 m achieves its maximum value of 0.88 when the confidence level is 0.335. The maximum *Precision* value is 1.00 when the confidence is 0.788. Maximum *Recall* is 0.97 when the confidence level is 0.0. The *F*_1_ score of improved YOLOv5m achieves its maximum value of 0.93 when the confidence level is 0.374. The maximum *Precision* value is 1.00 when the confidence is 0.696. Maximum *Recall* is 0.99 when the confidence level is 0.0. Therefore, by comparing with YOLOv3, YOLOv4, YOLOv5m, our improved model has better performance. As the *F*_1_ score increases, the proposed model’s *Precision* and *Recall* improve, and the model’s effectiveness improves significantly.

In addition, the *P-R* curve is further analyzed to evaluate the model performance. The *P-R* curves of different models are shown in Figure 10. In fact, the *P-R* curve is the *Precision-Recall* curve, where the *x*-axis represents *Recall*, and the *y*-axis represents *Precision*. The area enclosed under the *P-R* curve is *AP*, and the average value of *AP* of all categories is *mAP*. It can be observed from Figure 10 that the relationship between precision and recall draws a curve downward towards the right. From the *P-R* curve, one can observe that for YOLOv3, the maximum *mAP* is 0.927 when the threshold *IoU* is set to 0.5. For YOLOv4, the maximum *mAP* is 0.926 when the threshold *IoU* is set to 0.5. For YOLOv5m, the maximum *mAP* is 0.918 when the threshold *IoU* is set to 0.5. For improved YOLOv5m, the maximum *mAP* is 0.964 when the threshold *IoU* is set to 0.5. According to Figure 10, the area under the curve of the improved YOLOv5m is larger, and both precision and recall are high, which indicates that the model is robust.

Furthermore, we compared our improved model with the original YOLOV3, YOLOv4, and YOLOv5m on the same environment configuration. The experimental results are shown in Table 3. Table 3 shows that the *F*_1_ score of our model exceeds the original YOLOV3, YOLOv4, and YOLOv5m model’s *F*_1_ score. The *F*_1_ score is a comprehensive indicator of the robustness of the model. Hence, the proposed model is more robust for pest detection.

Table 3 demonstrates that our improved YOLOv5m is better than the original YOLOV3, YOLOv4, and YOLOv5m versions. The proposed model has several advantages in comparison to the original models. The comparison results in Table 3 show that even a small change to the deep model structure leads to a substantial improvement in the final performance relative to the original models. Additionally, these results also provide us with valuable insight into the potential of using the recommended model to identify pest insects in agriculture. This insight enables us to explore new and more powerful technology based on the mechanism of these models.

At the same time, we investigated the recognition efficiency of pest detection in several real environmental scenarios by using a CCD camera. Whereas, the detection accuracy of Figure 11 detected by the proposed YOLOv5 model is relatively high and can detect all pests from the labeled dataset. It reaches the upper 90%. Due to the influence of illumination and leaf overlap, there is a difference value between actual detection accuracy and theory detection accuracy.

To sum up, the improved model is significantly more efficient compared to the original YOLOv5m, YOLOv3, and YOLOv4. Based on our research, we found several studies comparing YOLOv5 to previous versions of YOLO, such as YOLOv4 or YOLOv3, as well as YOLOv5. According to a study provided by Li et al. [21], YOLOv5 was more efficient and about 2.5 times faster. YOLOv3-AFF was compared to YOLOv3, and *mAP* was increased by 10.18% [22]. Liu and Wang used improved YOLOv3 for tomato pests, improved YOLOv3 was compared to original YOLOv3, and detection accuracy was increased by 4.08% [23]. Liu et al. addressed an improved YOLOv4 to improve the accuracy of tomato pests, and the *mAP* (93.4%) of the improved YOLOv4 was better compared to YOLOv3 and YOLOv4 (73.4% and 87.1%, respectively) [24]. Zhang et al. introduced an improved YOLOv5 for unopened cotton boll detection in the field, the YOLOv5 had higher Average Precision than YOLOv5 and YOLOv3 [25]. YOLO-DPD was compared to YOLOv4 and YOLOv4 + RFA to detect and identify three diseases and pests on the rice canopy and the *mAP* of YOLO-DPD reached 92.24%. Comparing YOLOv4 and YOLOv4 + RFA, the *mAP* of YOLO-DPD was increased by 8.06% and 1.62%, respectively [26]. On the other hand, we encountered different studies that demonstrated that YOLO performs better than SSD in object detection. Guo et al. presented a YOLO-SIP detector to detect vegetable insects, and the *mAP* (84.22%) of YOLO-SIP was better when compared to YOLOv4 and SSD (67.88% and 8.76%, respectively) [27]. Better results and comparisons with other YOLO models were demonstrated by reference [28] while using jute pest identification model YOLO-JD for detecting pests from image. In our experiments, the modified YOLOv5m model is more robust and effective in detecting pests in scenes that require pest detection than the original models. Our improved YOLOv5m shows better results than YOLOv5m, YOLOv3, and YOLOv4. The experimental results demonstrate that the precision rate reached 95.7%, the recall rate reached 93.1%, the *mAP*@0.5 value reached 96.4%, and the *F*_1_ score reached 94.38%. Considering the previous discussion, the improved YOLOv5m is a wise choice for pest identification, and it would contribute to helping farmers to detect pests and improve the agriculture productivity quality.

## 4. Conclusions and Future Works

In this paper, four YOLO models are employed: YOLOv3, YOLOv4, YOLOv5m, and an improved version of YOLOv5m. Our work presents an improved YOLOv5m for detecting crop pests using images from the dataset with various resolutions. Three modifications have been made to the Yolov5m backbone, involving the SWin Transformer, ResSPP, and C3TR. The SWin Transformer (SWinTR) and Transformer (C3TR) mechanisms are introduced into the YOLOv5m network so that it can capture more global features and can increase the receptive field. They can increase the receptive field, which is to capture more global features. Furthermore, it is considered that ResSPP can enhance the network’s ability to extract more features from the backbone. In the feature fusion stage, three C3 output necks are converted to SWinTR in order to extract the global features from the feature map and forward them to the detection stage. Moreover, WConcat is added to the fusion feature, increasing the network’s feature fusion capabilities. Experimental tests demonstrate that the improved YOLOv5m is significantly better than the original YOLOv3, YOLOv4, and YOLOv5m models. The experiment results demonstrate that the precision rate achieved 95.7%, the recall rate achieved 93.1%, and the *F*_1_ score achieved 94.38%. In addition, the proposed algorithm achieved 96.4% *mAP* in relation to some state-of-the-art algorithms on the established dataset. Hence, the improved YOLOv5m could detect the targets with higher accuracy. Further work will be conducted to continue improving the network and designing the hardware in combination with related topics in order to provide possible opportunities for the application of the YOLOv5m model, which can be applied in a variety of scenarios, including the detection of pests from a variety of environments. Moreover, MobileNet may combine with YOLOv5m to improve detection accuracy.

## Figures and Tables

**Figure 1 insects-14-00054-f001:**
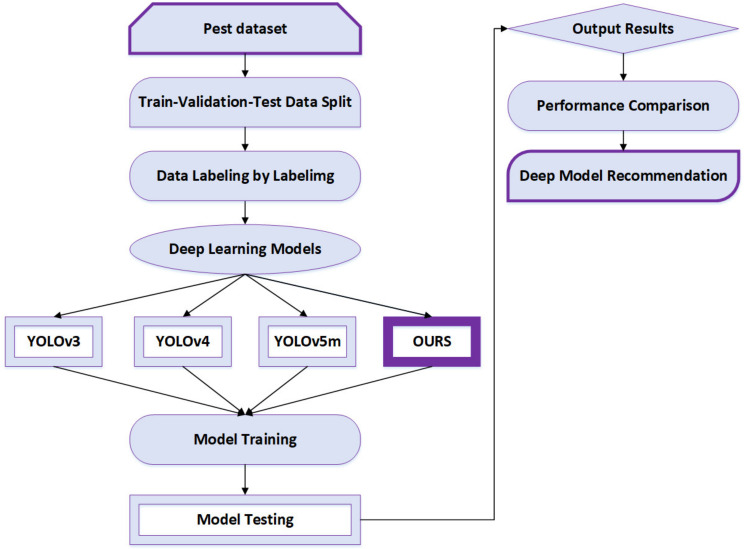
Overall flow chart of this work. It is composed of three important parts, i.e., pest dataset, deep learning model, and model testing. Pest dataset includes train, validation, and test data. Deep learning model considers various YOLO models. Model testing analyzes different performance indicators of algorithms for insect pest detection.

**Figure 2 insects-14-00054-f002:**
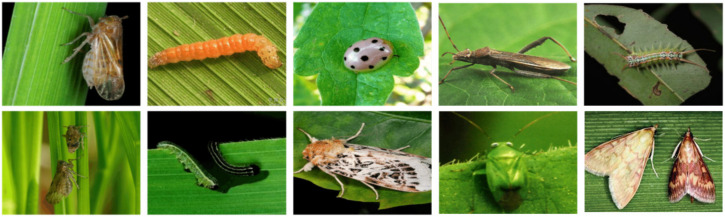
Sample of the pest dataset collected from the public datasets and online resources. The first and second rows display different kinds of pests and their resolutions, such as brown planthopper, rice leaf roller, ladybug, rice ear bug, caterpillar, moth, mirid bug, and corn borer.

**Figure 3 insects-14-00054-f003:**
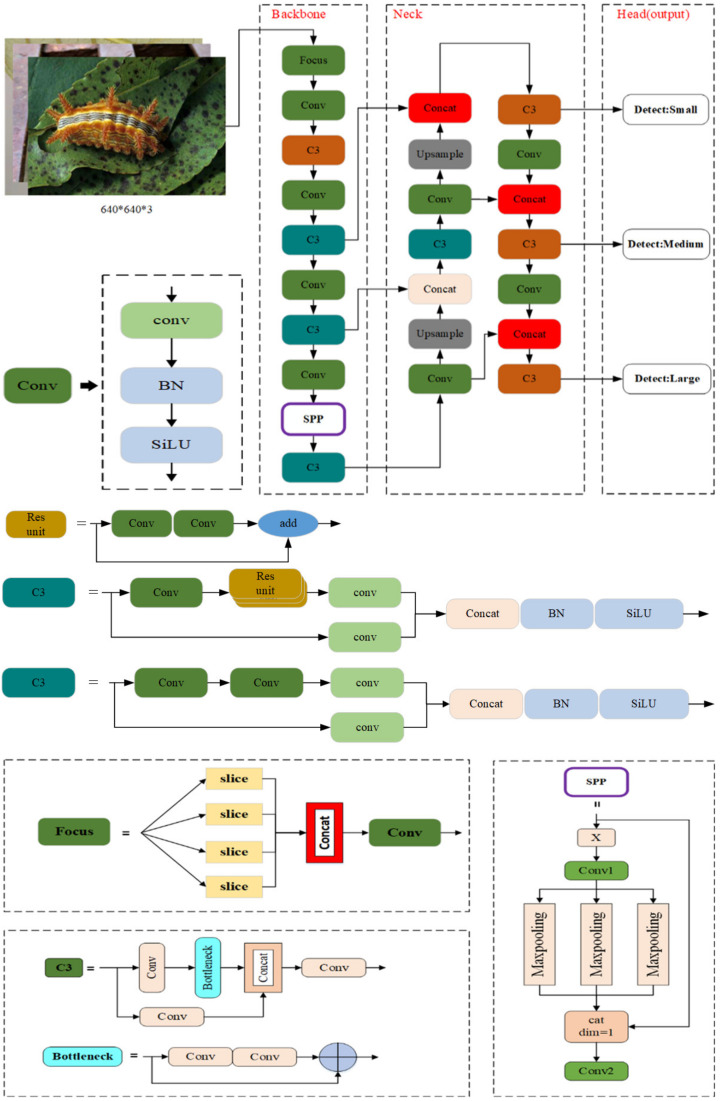
The structure of the YOLOv5m network. The backbone, neck, and output are its three main components. Feature extraction part of the YOLOv5m network includes Conv module, Focus module, C3 module, and SPP module.

**Figure 4 insects-14-00054-f004:**
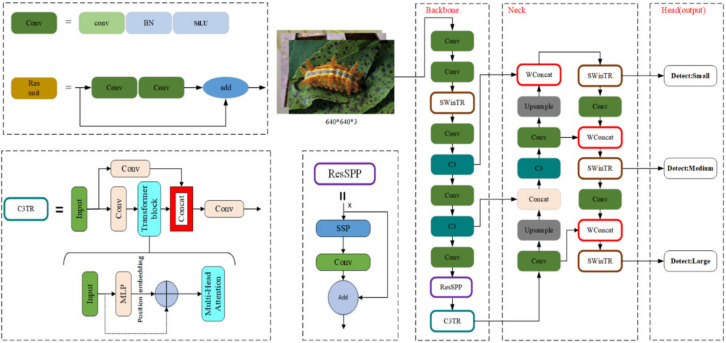
Improved YOLOv5m network structure. Backbone of the improved model includes Conv module, SWinTR module, ResSPP module, and C3TR module. SWinTR replaced C3 in the feature fusion part and added weights to the Concat.

**Figure 5 insects-14-00054-f005:**
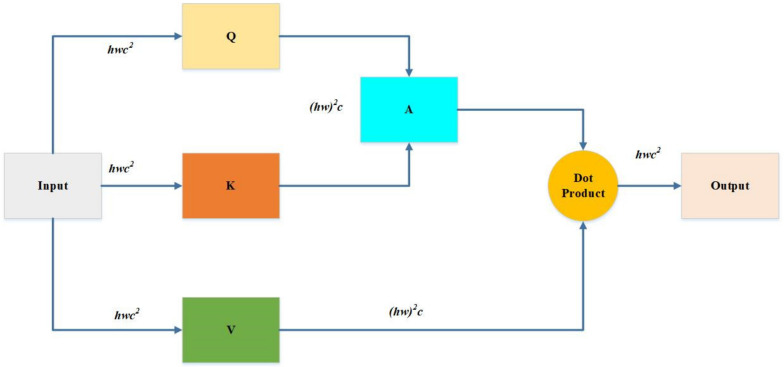
The computational cost of *MSA* and *W-MSA*. *MSA* is calculated according to Equation (3). *W*-*MSA* is calculated according to Equation (4).

**Figure 6 insects-14-00054-f006:**
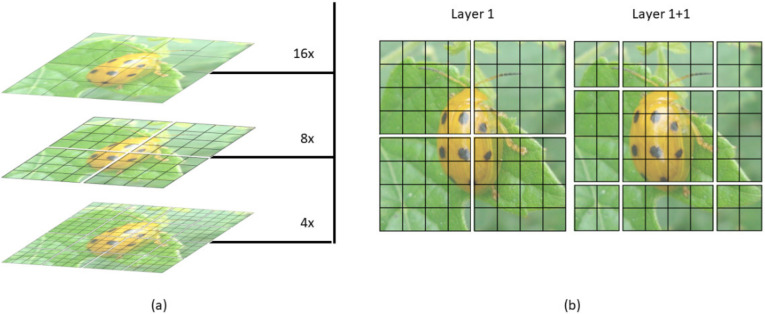
Illustration of (**a**) the hierarchical display and (**b**) the sliding window of SWin transformer (SWinTR).

**Figure 7 insects-14-00054-f007:**
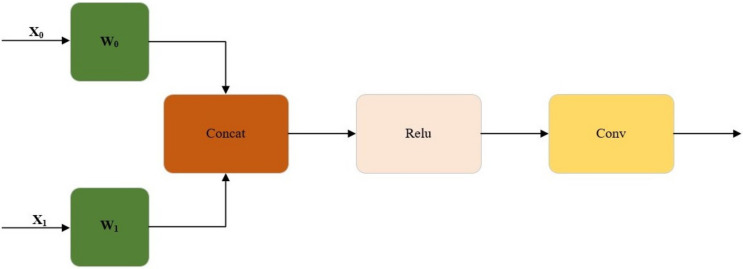
Overview of the WConcat splicing method. Feature maps x_0_ and x_1_ are spliced after W weight, then non-linearized with the Relu activation function, and then adjusted by convolution.

**Figure 8 insects-14-00054-f008:**
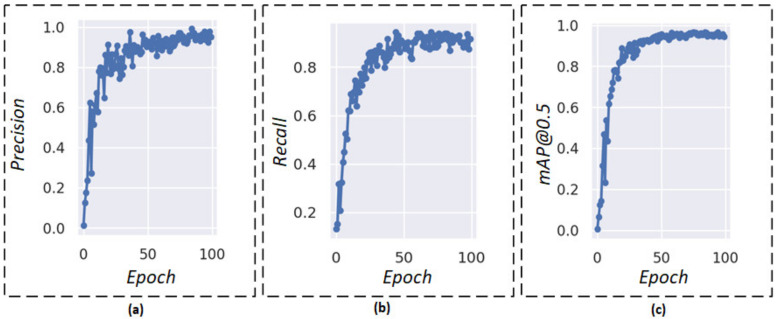
Evaluation indicators. (**a**) *Precision* result over epoch, (**b**) *Recall* result over epoch, and (**c**) *mAP* result over epoch.

**Figure 9 insects-14-00054-f009:**
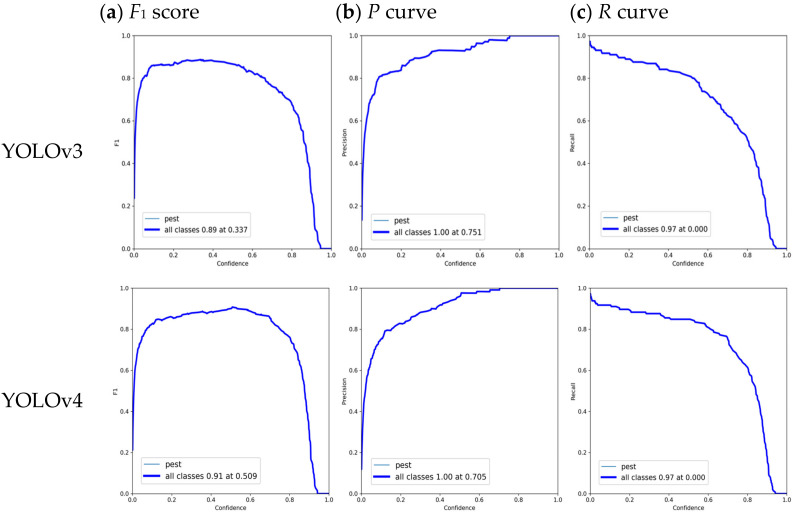
Comparison results of different YOLO models (YOLOv3, YOLOv4, YOLOv5m, and improved YOLOv5m) regarding performance indicators over confidence, including (**a**) *F*_1_ score, (**b**) *P* curve, and (**c**) *R* curve.

**Figure 10 insects-14-00054-f010:**
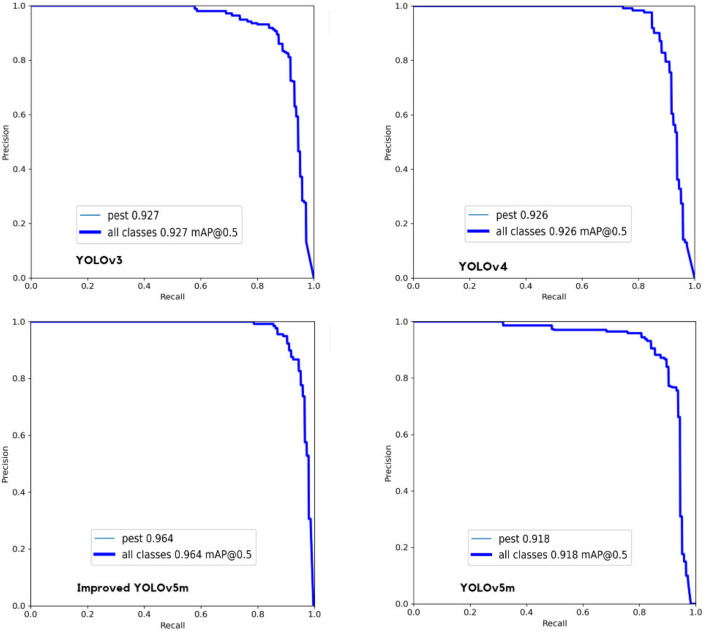
*P-R* curve of original YOLOv3, YOLOv4, YOLOv5m, and improved YOLOv5m, where the *x*-axis represents *Recall*, and the *y*-axis represents *Precision*.

**Figure 11 insects-14-00054-f011:**
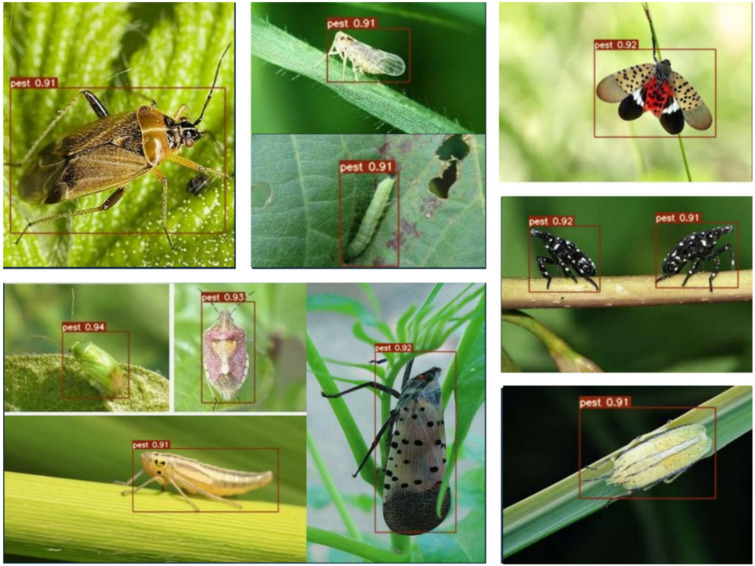
Detection accuracy of improved YOLOv5m for different insect pest images in real environmental scenarios such as field and greenhouse.

**Table 1 insects-14-00054-t001:** List of main abbreviations.

Abbreviations	Full Name
YOLO	YOU ONLY LOOK ONCE
SiLU	Sigmoid-Weighted Linear Units
SWinTR	SWin Transformer
C3TR	Transformer
Wconcat	Weight concat
MLP	Multi-Layer Perceptron’s
SPP	Spatial Pyramid Pooling
SVM	Support Vector Machine
KNN	K-nearest neighbor
Faster RCNN	Faster region with the convolutional neural network
SSD	Single shot multi-box detector
DPD	Diseases and Pests Detection
AFF	Adaptive feature fusion
SIP	Small Insect Pests
JD	Detecting jute diseases
CCD	Charge-coupled device
*mAP*	Mean Average Precision
*AP*	Average Precision
*MSA*	Multi-head Self-Attention Mechanism
*W-MSA*	Window-based Multi-head Self-Attention Mechanism
*ReLU*	Rectified Linear Unit
*P*	Precision
*R*	Recall
*IoU*	Intersection over Union

**Table 2 insects-14-00054-t002:** The experimental setting on algorithm parameter and its value.

Parameter	Value
Iterations	100
Batch size	10
Picture size	640 × 640
Learning rate	0.01
Momentum	0.937
Weight decay	0.0005

**Table 3 insects-14-00054-t003:** Comparison of experimental results for YOLOV3, YOLOv4, YOLOv5m, and proposed model with different performance indicators such as *Precision*, *Recall*, *mAP*, *F*_1_ score, and model size.

Model	*Precision*%	*Recall*%	*mAP*@0.5%	*F*_1_ Score%	Model Size/MB
YOLOv3	85.03	86.2	89.7	85.61	100
YOLOv4	87.83	89.63	91.87	88.72	117
YOLOv5m	88	86.9	90.35	87.44	40.4
Ours	95.7	93.1	96.4	94.38	38.1

## Data Availability

The data sets generated and analyzed in the present study may be available from the corresponding author upon request.

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
