# Peer review of "A New Pest Detection Method Based on Improved YOLOv5m"

_insects, 2023, doi:10.3390/insects14010054_

Round 1

Reviewer 1 Report

Dear authors,

Please find my comments directly added to the attached manuscript.

Author Response

(1) “The pest dataset used for YOLOv3, YOLOv4, YOLOv5m and an improved model for training and testing.” This sentence is a bit unclear. Do you mean that the created data set is used as input to the listed models? Did you use the same dataset for all these models?

Response: Thanks for the reviewer’s comments. Yes, we have checked that the created pest dataset is used as input to the listed models and the same dataset is used for all these models.

(2) I think that more information about the data set could be provided, for example about the background on which the pest are visible, the lighting conditions, etc. I am asking about that, as the background and the contrast can have an impact on the efficiency of the used algorithms. Do the pest (of the same type) have different sizes in the test images, are visible under different angles, etc.? For recognizing of how many pest types the algorithm was trained? How many samples per each type have been used? etc. In real (a commercial) application I suppose that learning data set with c. 1300 may be insufficient. To achieve a high efficiency even two orders of magnitude more data samples may be required. I know that it is not easy to collect such a data set, therefore it it is not a problem in this work, but for the future you could consider creation of artificially modified samples (rotated, on replaced background, etc.)

Response: Thanks for the reviewer’s comments. We have provided more information about the data set in our revised version. We have eliminated the influence of background on the algorithm by using image processing techniques like PhotoCopy algorithm and GrabCut algorithm. The pest of each category has the same size in the test images and pests are visible under different angles. Ten pest types were trained for the algorithm and each type shared together 1309 images from dataset, of which 1178 were used for training and 131 for validation and test shared together. As the reviewer said, the learning data set may be insufficient. Hence, we will collect more data set and consider creation of artificially modified samples in future work.

(3) Could you provide more details about the structure of particular layers in (1 - Backbone) and (2 - Neck)? For example, in (1) you perform several convolution operations. What kind of masks did you apply at each of these stages? What is performed in the C3 layers? Do the C3 layers also include downsampling operation? I am asking as in (2) again you have the convolution layers alternately used with upsampling and the Concat operations. The question is if the Concat blocks require images with the same sizes. Such details would be very valuable to better understand the structure of your system.

Response: Many thanks. We have provided more information about the structure of the Backbone and Neck in the revised version. We have added two C3 layers for better understanding in Figure 3, one is for the Backbone, and the other is for Neck. C3 contains two blocks, one is convolution, and the other is Resnet. There is also convolution in Resnet. In Figure 3, the structure of C3 can be seen in detail. We do not have any downsampling operations in C3. Our upsample has been used alternately with the concat. When the concat is performed, we require the size of our feature map to be the same for evaluating the image. To ensure that the two feature maps are consistent, two approaches could be considered. First, we have upsampled a relatively small feature map and then turned it into a relatively large feature map, then fused it with other large feature maps, which is an upsampling operation. Furthermore, for the down-sampling operation, a relatively large feature map becomes a relatively small feature map, and then the small feature map is concat with another smaller feature map. In the feature synthesis part, there are two operations, one is up-sampling, and the other is convolution compensation. The convolution compensation plays the role of down-sampling.

(4) Did you compare the computation complexity of the used methods?

Response: Thanks a lot. At present, we have focused on the accuracy of the algorithm performance. Hence, we will consider the computation complexity of the used methods in further work.

(5) How the downsampling is performed? Which downsampling factors are used, and what kind (if any) of ow pass filter kernels are applied to avoid aliasing?

Response: Thanks for the reviewer’s comments. In fact, the downsampling operation was performed based on the Pooling layer along the spatial dimensions. According to specific mathematical operations such as maximum pooling, the downsampling layer (also known as the pooling layer) can reduce the spatial dimensions of the image. In our manuscript, the stride of the network was used as downsampling factors. The stride of any layer in the network is equal to the factor by which the layer's output is smaller than the input image to the network. Owing to our limited consideration, we have not applied ow pass filter kernels to avoid aliasing.

(6) “The Transformer's self-attention mechanism can effectively pick up global features.” What are the global features? Do you mean such features as shape, size, a general color, etc.?

Response: Thanks for the reviewer’s comments. Yes, as the reviewer said, the global features described such features as shape, size, a general color, etc.

(7) “then the window number is h/w × w/M. ” This sentence is unclear. First you divide the feature map (what is its example size?) into smaller windows, and then each window is further divided into MxM patches. With a given number (MxM) of patches and the map sizes of h x w do you compute the total number of windows in the feature map? The h/w x w/M term is not clear.

Response: Thanks a lot. There is a mistake. Since the map size is h×w, the window number is h/ M × w/M.

(8) Since you mention the area as the indicator (the larger -> the better) of the quality, perhaps a good option would be to provide the values in the white areas of the graphs (now empty). At the first sight, the performance of the "improved YOLOv5m" is not the best one (a smaller area than in case of the not improved case). Then below the Figure 9 you provide more details on the performance analysis. I think that the provided values could be highlighted in the Figure, to facilitate the analysis. For example, you mention the confidence level. What this value tells us about the precision? In general, how the provided values of Recall and Precision correlate with the "area" factor.

Response: Thanks for the reviewer’s comments. We have provided the values in the white areas of the graphs to show the performance indicators in new Figure 9, where the area enclosed under the PR curve is AP, and the average value of AP of all categories is mAP. The relationship between precision and recall draws a curve downward towards the right. When the area under the curve is larger, both precision and recall are high, which means the model is robust. According to Figure 9 we can see that the improved YOLOv5m is robust. In addition, according to Table 3, the confidence threshold increases, the precision goes up and the recall goes down. Also, it can be seen that the F1 curves confidence value optimizes the precision and recall. The F1 score curve represents the balance between precision and recall. In many cases, a higher confidence value is desirable.

Reviewer 2 Report

Paper: insects-2025971

Title: A New Pest Detection Method Based on Improved YOLOv5m

Research aim:

The present research main question is to improve a YOLO models for pest detection in plant based on a deep convolutional neural network, the topic is very important and it is relevant in the field, certainly it would contribute to help farmers to detect pest and improve the agriculture productivity quality.

General comments:  

Abstract:

There is a lot of redundancy, the same idea about “importance and high accuracy and precision of authors models in pest detection” repeated several times with different sentences

Introduction:

1.      Please check manuscript sections parts organisation from line 115 to 119

2.      Same remark as abstract, there is a lot of redundancy, please try to improve the introduction text.

Materials and Methods:

1.      Add  all equation reference

2.      Line 138  change figure 2 instead figure 1

3.      Line 210 and 211  query, key and weight value

Experimental results and analysis

1.      Reword figure 8 title and please use letter (a, b and c) instead left, right and middle  (line 360-361)

2.      Line 376, “Figure 9 shows that the original YOLOv3 F1 score…..when the confidence level………..”  How did authors find F1 score and confidence level values and what its relationship with figure 9 which present model Precision versus Recall for each YOLO version.

3.      Results not clear

4.      Table 3 data, how authors can explain why the YOLOv3 value are better than those of YOLOv4 and YOLOv5m.

5.      Absence of the discussion part, authors should discuss their results according to those in literature.

Author Response

(1) Abstract: There is a lot of redundancy, the same idea about “importance and high accuracy and precision of authors models in pest detection” repeated several times with different sentences.

Response: Thanks for the reviewer’s comment. We have improved the abstract. The repeated expression has been deleted in the revised version.

(2) Introduction: Please check manuscript sections parts organisation from line 115 to 119

Response: Thanks a lot. We have checked each section. According to the architecture of the manuscript sections parts, the title of section 4 “Conclusions” has been replaced with “Conclusions and future works”.

(3) Introduction: Same remark as abstract, there is a lot of redundancy, please try to improve the introduction text.

Response: Thanks a lot. According to reviewer’s suggestions, we have improved the introduction section in the revised version. A lot of redundancy has been deleted.

(4) Materials and Methods: Add all equation reference

Response: Thanks a lot. We have added references for Eqs.(1)-(6) in the revised version. Eq.(7) and Eq.(8) were proposed by us for better understanding and they did not need to add any reference.

(5) Materials and Methods: Line 138 change figure 2 instead figure 1

Response: Thanks a lot. The minor mistake has been corrected.

(6) Materials and Methods: Line 210 and 211 query, key and weight value

Response: Many thanks. The minor mistake has been corrected.

(7) Experimental results and analysis: Reword figure 8 title and please use letter (a, b and c) instead left, right and middle (line 360-361)

Response: Many thanks. According to the reviewer’s good advice, Figure 8 title changed to (a, b and c) instead left, right and middle.

(8) Experimental results and analysis: Line 376, “Figure 9 shows that the original YOLOv3 F1 score…..when the confidence level………..”  How did authors find F1 score and confidence level values and what its relationship with figure 9 which present model Precision versus Recall for each YOLO version.

Response: Thanks for the reviewer’s comment. We have added Table 3 to explain F1 score, P curve, and R curve over confidence for different YOLO models in the revised version. The F1 score curve represents the balance between precision and recall. We have provided the values in the white areas of the graphs to show the performance indicators in new Figure 9, which could explain the relationship clearly.

(9) Experimental results and analysis: Results not clear

Response: Thanks a lot. We have redescribed these results in the revised manuscript.

(10) Experimental results and analysis: Table 3 data, how authors can explain why the YOLOv3 value are better than those of YOLOv4 and YOLOv5m.

Response: Thanks a lot. Because of our lack of rigor, the data of YOLOv3 was recorded in the YOLOv4 row while the data of YOLOv4 was recorded in the YOLOv3 row. We have adjusted the order of the data.

(11) Experimental results and analysis: Absence of the discussion part, authors should discuss their results according to those in literature.

Response: Thanks a lot. According to reviewer’s suggestions, we have added the discussion description in the revised version.

Round 2

Reviewer 1 Report

Dear Authors,

After the corrections, the paper looks much better. I currently have no new comments on the proposed solutions.

What I also noticed is some confusion in the way of notation of signals, variables, etc. For example, in equation 11 you use AP and mAP variables. They are written in italics in these equations. At the same time, in the content of the paper, these variables are written in a plain font (lines 361, 374, 378, ...). AP a few lines below is written in a much larger font than the rest of the text, which does not look good. In equation 12 you use the variables Precision and Recall, which are again in italic font here, and in the text (line 351) they are in the plain font. The F1score variable is similarly written differently in the formula and differently in the text. It doesn't look good. The form of expressing variables should be standardized for all variables used in the article.

Author Response

Thanks for the reviewer’s comments. We have unified the form of expressing variables in the revised version.

Reviewer 2 Report

 in line 390, authors should use figure (X)  instead table 3, because authors presents figure not numbers 

Author Response

Many thanks. We have used figure 9 instead table 3 in the revised version.